# Considerations for conducting psychological research in lower- and middle-income countries

Samia C. Akhter-Khan, Sakshi Ghai & Rosie Mayston

Beginning to conduct psychological research in lower- and middle-income countries (LMICs) is daunting. In this reflexive commentary, the authors raise three critical questions that researchers should ask themselves before conducting research in LMICs.

For over a decade now, it has been established that more psychological research needs to include samples from lower- and middle-income countries (LMICs)[1]. Scholars have also pointed out the many challenges that are involved in doing research in low-resource settings, including restricted funding, ethical considerations, and power imbalances, such as adequately reflecting collaborations with local partners in academic authorship[2]. Here, we—two early-career psychologists from Germany and India and one global health researcher from the UK who all conducted field work in LMICs—summarize key questions and concerns that we wish someone would have told us before conducting our research in Ethiopia, Myanmar, Thailand, and India. While there is a need for more psychological research in LMIC, we need to proceed with caution and with the goals of trying to contribute to capacity strengthening and benefiting communities.

## Why am I the right person to conduct this particular research?

Doing good cannot be uncritically assumed in global health or psychological research. It is essential that we, individuals in privileged positions, question our motivations for wanting to work with people who are disadvantaged (e.g., in terms of socio-economic status, ethnicity, or health). As researchers we operate in systems which reward individual achievements and intellectual gains: what can we do to ensure our work benefits the communities who participate in research and avoid pursuing practices which are knowingly or unknowingly exploitative? Research should emerge from genuine need and real relationships with community members. For instance, when working with vulnerable community members, such as low-income older people, key concerns of participants will be how to meet basic health needs. Responding to these needs (e.g., by getting participants glasses so they can read the consent forms they are filling out) may go beyond the proposed research and funding guidelines but needs to be considered prior to planning the project.

Instead of leading research to uncover truths about the lives of foreign others, the research endeavour should be framed as a partnership between local communities and external researchers. External researchers' primary role should be as a conduit for technical expertise, material resources, and capacity strengthening, rather than owning a singular intellectual achievement. In other words, within the constraints of a rigged system, an outsider's primary motivation should be making incremental contributions to equity[3].

It's important to remember that benefits do not just come from research outputs; in Ethiopia, the most tangible impacts of Rosie's work have been from contributions to mentoring and training, supporting early career researchers to successfully lead and publish peer-reviewed publications, apply for PhD fellowships, and transition into academic roles. We have tools to support meaningful engagement: participatory approaches to research offer frameworks for engaging with local communities, to help us to understand their problems, priorities, values, and preferences[4]. Genuine relationships with local partners––whether these are academics, staff from non-governmental organisations, civil society organisations, community and religious leaders––should be established before implementing a project to set an essential foundation for participatory research with communities. Ensuring that the contribution of partners to projects are appropriately recognised, whether this is via compensation for time spent, support for skills development, and authorship of academic papers, is another core principle of ethical work[5]. There remains a mountain to climb on this most basic tenet of equity: a third of studies included in our systematic review of qualitative studies about experiences of loneliness in LMIC were authored solely by those at higher-income country (HIC) institutions[6]. It is important to note that far from being incompatible with high quality outputs, complex multisectoral partnerships focussed on equity make for better research and better experiences for researchers, enriched as they are by debate and occasional discord, collaboration, and friendship.

## Do I have time for slow research?

Once the relevance of the research question to the local community and the work relationship with local collaborators has been established, the slow, iterative process of familiarisation, flexibility, and building trust can begin.

**Familiarisation**. Working in a new setting requires time to familiarise oneself with the research question and context. Most countries are linguistically and culturally diverse. The preference to speak in one's native language may especially be the case for psychologically sensitive topics (e.g., mental health), which is why local partners or at the very least, translators (including for local dialects) are essential. There may be limited published work on the topic of interest to begin with, given the sparse number of academic publications in psychology from LMICs[1]. Thus, it is often important to first conduct qualitative research to explore local understandings and expressions of key concepts. These cannot be assumed to be universal, for example, the prominence of different symptoms related to depression varies regionally and is not completely aligned to clinical diagnostic criteria from HICs[7]. Formative work will help to ascertain how a meaningful research goal might be (co)developed.

**Flexibility**. Even when a thorough research plan has been established and a topic guide was co-created with local experts, there may be methods or

questions that will simply not work and need to be iteratively adjusted. In Southeast Asia, a common concept termed kreng jai (เกรงใจ) in Thai (anade (အားနာတယ်) in Burmese, phiền in Vietnamese), which means to not want to burden others (e.g., by sharing personal problems and making others feel bad), prevented people from opening up to others in focus group discussions[8]. Hence, other research methods that involve ethnographic approaches to establish trusting relationships with participants may be more adequate in certain contexts. Moreover, flexibility is needed when working in rural settings with limited infrastructure (e.g., power shortages, limited internet), restricted areas (e.g., waiting for permits, longer travel time), or political instability. For instance, research projects in Myanmar had to be cancelled and moved to Thailand following the coup d'état in 2021, as international funders would not send grant money to Myanmar.

**Trust**. Conducting research with local people requires establishing trust: between the researcher and participants, interpreters, and collaborators. Trusting relationships take time, especially in countries affected by recent conflict[9]. The TRUST code which stands for trust, fairness, respect, care, and honesty has been routinely advocated to promote equitable research partnerships[2]. However, the TRUST code also needs to be adapted to account for unique challenges that emerge when working in countries where there are very limited existing research guidelines. When Samia began her qualitative work on loneliness in Myanmar in 2019, following over 60 years of military rule, one of her participants had security concerns after the interview as he had spent several years in the military himself (which led to deleting the interview recording). Learning from this experience, Samia made sure to explicitly promise not to discuss the political situation in the country when introducing her participatory project with Myanmar migrants in Thailand in 2023. By ensuring on-going communication with participants, she was able to put them at ease and build trust, which in turn made them more open to contribute[10]. This example illustrates how even psychological concepts such as loneliness can be perceived as political in a conflict-affected setting, requiring extra sensitivity in the research endeavour.

## How can I implement the TRUST code into my research practice?

To truly ensure a value-based approach to conducting research in diverse settings, it is critical to follow ethical principles tailored to the local context, embed reflexivity, and build respect in research partnerships.

**Apply ethical guidelines in context**. To avoid unethical research practices (e.g., ethics dumping), it is critical to question the *lens* we use to shape and inform ethical guidelines. While seeking informed consent is non-negotiable, we argue that ethical processes should be carefully embedded in local sensitivities. This involves building equitable local partnerships to understand best practices for seeking consent and co-production of tailored consent processes. For example, in addition to formal ethics approval, our research required informal channels of consent from village leaders in Thailand and from family units in India.

**Embedding reflexivity**. Whether it is qualitative or quantitative research, it is important to question our subjectivity as researchers and be aware of preconceptions that are formed by psychological models taught at HIC institutions. For example, researchers may have preconceptions that loneliness does not exist in socially embedded cultures because these are considered to be more collectivist in nature. This might lead to inadvertent romanticisation of the social lives in LMIC. Researchers who grew up in LMICs and who were trained partly or completely in Western systems/ paradigms are also often far removed from research participants: economically, culturally, politically[11]. Researchers conducting fieldwork in rural areas of LMICs may experience a divergence between their own background and the societal norms prevalent in the context. For example, as an unmarried, educated female researcher, Sakshi differed significantly from the societal expectations of the North-Indian rural contexts, where the gender norms may include expectations of women being married, and primarily focused on household roles over pursuing higher education or professional careers. Such differences—whether related to education, marital status, or professional roles—will introduce inevitable power dynamics, bringing about a heightened need for reflexivity. While we cannot entirely mitigate the potential of a researcher's identity influencing the research process, we can reflect on and question our own privilege in the context we are working in.

**Building respect in research partnerships**. There is a fundamental need to respect the *dignity* of participants in LMICs. Respect in research partnerships goes beyond general ethical principles and involves actively listening to others' perspectives, being prepared to learn, and ceding power where necessary. In LMICs, this can include adapting research methods, agreeing on meeting times and locations that accommodate local stakeholders, and being mindful of the differing levels of time and capacity people have for research. Moreover, people with lived experience of mental health conditions, which are often stigmatised and present significant challenges for individuals and their families, may find it particularly difficult to participate in research that seeks their experiences and perspectives[12]. Demonstrating respect in research endeavours requires acknowledging the unique insights these individuals offer and adapting methods to ensure inclusion, rather than defaulting to more accessible populations. By doing so, we can foster a research process that prioritises dignity and inclusivity.

## Conclusion

Conducting psychological research in lower-resource settings requires researchers, particularly those based in the Global North, to critically reflect on their own motivations and suitability for research in the Global South. Working in LMICs has made us actively reflect on how we can use our privileged role as researchers from HICs (or affiliated with institutions in HICs) to make a positive impact for the communities we work with, including in HIC contexts. It is relatively easy as a researcher in the Global North to become successful by branching out into more global research, justifying this as altruism, a motivation to address inequity. Over time and with increased knowledge of the historical, economic, and social context in which this takes place, it has become a much bigger challenge to explain why me, why this work? We have a shared recognition of how much we have personally gained from the kindness and openness of others to collaborate and share their knowledge and experiences. Whilst the impact of research can be difficult to ascertain, we hope that our shared approach–reflexive, flexible, focussed upon the relevance of our work (and ways of working)–has led to incremental reciprocal benefits for the people we have worked with, including mutual learning, skills and career development. It is crucial to carefully consider how to engage ethically and collaboratively with partners in low-resource settings, ensuring that research approaches are respectful and contextually appropriate. Centring the rationale for any research endeavour around the interests of the local community is of utmost importance, specifically where resources are sparse and basic needs are not met. Altogether, the challenges mentioned in this commentary speak for the importance of building long-term research partnerships and spending enough time in the country where the research is to be conducted, ideally

before (co)designing and implementing a project. Only then can one begin to understand how to contribute research that is meaningful and valuable to local communities, in an ethical way that preserves dignity and builds resources to address problems, rather than being extractive.

**Samia C. Akhter-Khan** ⓘ[1] ✉**, Sakshi Ghai[2] & Rosie Mayston[1]**

[1]Department of Global Health & Social Medicine, King's College London, London, UK. [2]Department of Psychological and Behavioural Science, London School of Economics, London, UK.
✉e-mail: Samia.akhter-khan@kcl.ac.uk

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

### Author contributions

S.A.: Conceptualization, Writing—Original Draft, Writing—Review & Editing. S.G.: Writing—Original Draft, Writing—Review & Editing. R.M.: Writing—Original Draft, Writing—Review & Editing.

### Competing interests

The authors declare no competing interests.
