## [Transparent Peer Review file · Communications Psychology]

Considerations for conducting psychological research in lower- and middle-income countries

Corresponding Author: Ms Samia Akhter-Khan

Version 0:

Decision Letter:

**** Please ensure you delete the link to your author homepage in this e-mail if you wish to forward it to your co-authors ****

Dear Ms Akhter-Khan,

Your Comment titled "Conducting psychological research in lower- and middle-income countries – where to start?" has now been seen by 3 referees, whose comments appear below. In light of their advice, I am delighted to say that we are happy, in principle, to publish it in Communications Psychology.

We will not send your revised paper for further review if, in the editors' judgment, the referees' comments on the present version have been addressed. If the revised paper is in Communications Psychology format, in an accessible style, and of appropriate length, we shall accept it for publication immediately. I have attached an edited version of your manuscript, and ask you to attend to each comment in detail.

EDITORIAL REQUESTS:

* Please review the changes in the attached copy of your manuscript, which has been edited for style, and address the comments and queries I have added. If using Word, please use the 'track changes' feature to make the process of accepting your manuscript more efficient.

* Communications Psychology uses a transparent peer review system. On author request, confidential information and data can be removed from the published reviewer reports and rebuttal letters prior to publication. If you are concerned about the release of confidential data, please let us know specifically what information you would like to have removed. Please note that we cannot incorporate redactions for any other reasons.

*If you have not done so already, please alert me to any related manuscripts from your group that are under consideration or in press at other journals, or are being written up for submission to other journals (see www.nature.com/authors/editorial_policies/duplicate.html for details).

SUBMISSION INFORMATION:

* If you wish, you may also submit a visually arresting image, together with a concise legend, for consideration as a 'Hero Image' on our homepage. The file should be 1400x400 pixels and should be uploaded as 'Related Manuscript File'. In addition to our home page, we may also use this image (with credit) in other journal-specific promotional material.

In order to accept your paper, we require the following:

* A cover letter describing your response to our editorial requests.

* A separate document detailing your point-by-point response to any issues raised by our referees (please include the referees' comments in this document).

* The final version of your text as a Word or TeX/LaTeX file, with any tables prepared using the Table menu in Word or the table environment in TeX/LaTeX and using the 'track changes' feature in Word.

* Production-quality versions of all figures, supplied as separate files. Photographic images should be 300 dpi in RGB format (.jpg, TIFF or native Photoshop format) and any labels/scale bars included in a separate layer from the image. Line art, graphs and schemes should be vector format (.ai, .eps, .pdf); Adobe Illustrator files are preferred and will minimize production time. Any chemical structures or schemes contained within figures should additionally be supplied as separate Chemdraw (.cdx) files.

Communications Psychology is a fully open access journal. Articles are made freely accessible on publication. For further information about article processing charges, open access funding, and advice and support from Nature Research, please visit <https://www.nature.com/commpsychol/open-access>

At acceptance, you will be provided with instructions for completing the open access licence agreement on behalf of all authors. This grants us the necessary permissions to publish your paper. Additionally, you will be asked to declare that all required third party permissions have been obtained.

Please note that your paper cannot be sent for typesetting to our production team until we have received this information; **therefore, please ensure that you have this ready when submitting the final version of your manuscript.**

ORCID

Communications Psychology is committed to improving transparency in authorship. As part of our efforts in this direction, we are now requesting that all authors identified as 'corresponding author' create and link their Open Researcher and Contributor Identifier (ORCID) with their account on the Manuscript Tracking System (MTS) prior to acceptance. ORCID helps the scientific community achieve unambiguous attribution of all scholarly contributions. For more information please visit <http://www.springernature.com/orcid>

For all corresponding authors listed on the manuscript, please follow the instructions in the link below to link your ORCID to your account on our MTS before submitting the final version of the manuscript. If you do not yet have an ORCID you will be able to create one in minutes.

IMPORTANT: All authors identified as 'corresponding author' on the manuscript must follow these instructions. Non-corresponding authors do not have to link their ORCIDs but are encouraged to do so. Please note that it will not be possible to add/modify ORCIDs at proof. Thus, if they wish to have their ORCID added to the paper they must also follow the above procedure prior to acceptance.

To support ORCID's aims, we only allow a single ORCID identifier to be attached to one account. If you have any issues attaching an ORCID identifier to your MTS account, please contact the [Platform Support Helpdesk](http://platformsupport.nature.com/).

Link Redacted

We hope to hear from you within two weeks; please let us know if the process may take longer.

Best regards,

Jennifer Bellingtier

Jennifer Bellingtier, PhD
Senior Editor
Communications Psychology

REVIEWERS' EXPERTISE:

Reviewer #1 Research in LMICs
Reviewer #2 Research in LMICs
Reviewer #3 Research in LMICs

REVIEWERS' COMMENTS:

Reviewer #1 (Remarks to the Author):

Topic: Very important and of interest to HIC Global North researchers wanting to conduct research in LMICs. The concerns addressed through 3 sections (suitability of the researcher to pursue that study, patience with, the often 'slow', research process and capturing participant trust) are legitimate in that they make up important recurring concerns of researchers coming into LMICs. The topic is particularly important to researchers in the psychological sciences and even to generally other research fields.

To improve the work: This work needs a lot of improvement to be accepted for publication.

1. In line 23 authors say that for over a decade now it has been established that there is lack of psychological research in low income settings. There is a difference between low income settings and LMICs which is the main focus of this work. Plus it is inaccurate to state that there is no such research in these settings as lots of research projects can be accessed through google or other search functions.
2. In the second sentence, please add more details by what you mean by ethical considerations, restricted funding that you refer to. I see it's a quotation but for the reader your own description is required to make this quote make your work clearer.
3. Name the research site in India as you have named the site in Thailand.
4. Sentence in line 28-29 about psychologists needing to learn from other disciplines seem misplaced unless more context is provided. Eg What is it they do not know? What can other disciplines add to what psychologists know?
5. First question is about researcher's suitability to conduct the study. I think it will be clearer if the authors give examples from their experience in India and Thailand eg their qualifications and its suitability for the study similar to what they do in discussing the other two questions. Additionally, I think it's a fallacy to imagine that participants in LMICs are 'disadvantaged' as the author mention in this section. OR they can explain what they mean by disadvantage. Also explain what they mean by 'doing good'. Not all

Reviewer #2 (Remarks to the Author):

This paper reflects on and acknowledges the challenges associated with conducting psychological research in lower-and middle-income countries (LMICs), based on the authors' research experiences in Myanmar, Thailand and India. The authors then proceed to pose and answer three critical questions that they believe researchers seeking to conduct or conducting psychological research in LMICs should be aware of, and reflect upon, to avoid potential pitfalls, and being exploitative. The paper builds upon neo-colonial science/helicopter research literature, and makes important practical contribution in terms of conducting responsible, ethical and decolonial research, particularly those involving participants from the Majority World settings. The contribution will be beneficial to both researchers and practitioners, and has the potential to influence thinking in the field.

There are a few minor concerns I wish to bring to the attention of the authors for consideration:

1. Line 30: Take another look at this sentence, the last part seems a bit awkward for me. Do you mean "...trying to contribute to capacity strengthening to (instead of 'and') benefit communities?"
2. Line 33: Check the first question again. The content of the response doesn't seem to align very well with the question posed here. It seems to me that the question should rather be "Why do I need to involve/include local partners in this particular research?" or something similar.
3. Line 48: It may be helpful to qualify/define the relationship here. Genuine relationship is preferred. This is important because of the various shapes and forms of unethical parachute/helicopter research that seem to be gaining traction the the academy. E.g., what qualify as local involvement? For e.g., when privileged academics from Minority World contexts/Global North pre-design every aspect of a research project, and only seek a couple of local partners to include for purposes of box-checking, they may think they are engaging in participatory research, because it involves relationship with local partners. However, this may not qualify as ethical and genuine research collaboration/relationship.

Line 52-53: Please clarify whether these are either or situations. For example, is it enough to provide monetary compensation to a local RA without consideration for authorship? Should we even move to (would you advocate) crediting local participants with authorship, given that, as academics, we benefit from their rich expertise and knowledge- what we call data?

Line 70: 'there' should be 'There'

Line 75: 'formative' should be 'Formative'

Line 92-93: "semi-democratic" Myanmar? Whose evaluative definition is this? You or the local partners/participants? What does it mean? I would avoid such value-laden and evaluative terms.

Line 147: The thought about spending enough time in an LMIC where research is to be conducted, preferably before co-designing and implementing a project, is a good point- should be mentioned earlier as well (see comment 3, line 48). Do you have ideas about how much time is 'enough', based on your own individual experiences?

Regards
Stephen Baffour Adjei

Reviewer #3 (Remarks to the Author):

The manuscript addresses the ethical and practical challenges of conducting psychological research in low- and middle-income countries (LMICs), specifically Myanmar, Thailand, and India. The authors summarize key questions and concerns that researchers should consider in these contexts, emphasizing the importance of ethical reflexivity, community engagement, and culturally sensitive methodologies.

While the manuscript raises relevant points, much of the content overlaps with established ethical frameworks, such as the TRUST Code, and existing guidelines from various journals, including Nature. The repetition of familiar ethical principles may lead to a perceived lack of novelty. The authors state in the introduction, "Here, we summarize key questions and concerns that we wish someone would have told us before conducting our research in Myanmar, Thailand, and India." As readers, we expect to learn firsthand information from their experiences in different LMIC settings. However, upon reviewing the text, we find that much of the content aligns with well-established ethical frameworks.

The authors provide some examples from their experiences, which is commendable. However, I believe they could enhance their contribution by offering more detailed accounts of specific challenges faced during their research in the three countries, particularly how these experiences shaped their methodologies and ethical considerations. This could provide valuable insights for researchers encountering similar situations.

Furthermore, including a critique of existing ethical guidelines, including the TRUST Code, could illuminate areas needing improvement and spark a dialogue on how these codes can be adapted for diverse contexts. Overall, while this manuscript serves as a valuable starting point, it has the potential to offer more original insights for researchers in the field.

Reviewer #1 (Remarks to the Author):

Topic: Very important and of interest to HIC Global North researchers wanting to conduct research in LMICs. The concerns addressed through 3 sections (suitability of the researcher to pursue that study, patience with, the often 'slow', research process and capturing participant trust) are legitimate in that they make up important recurring concerns of researchers coming into LMICs. The topic is particularly important to researchers in the psychological sciences and even to generally other research fields.

Response: We want to thank the author for their positive evaluation and helpful comments.

To improve the work: This work needs a lot of improvement to be accepted for publication.

1. In line 23 authors say that for over a decade now it has been established that there is lack of psychological research in low income settings. There is a difference between low income settings and LMICs which is the main focus of this work. Plus it is inaccurate to state that there is no such research in these settings as lots of research projects can be accessed through google or other search functions.

Response: We have changed low-income settings to LMIC and now write: "For over a decade now, it has been established that more psychological research is needed in lower- and middle-income countries (LMICs)".

2. In the second sentence, please add more details by what you mean by ethical considerations, restricted funding that you refer to. I see it's a quotation but for the reader your own description is required to make this quote make your work clearer.

Response: We changed the sentence accordingly: "Scholars have also pointed out the many challenges that are involved in doing research in low-resource settings, including restricted funding, ethical considerations, and power imbalances, such as adequately reflecting collaborations with local partners in academic authorship"

3. Name the research site in India as you have named the site in Thailand.

Response: We are not sure what this comment refers to, as we do not name a research site in Thailand (we only name countries: "Ethiopia, Myanmar, Thailand, and India").

4. Sentence in line 28-29 about psychologists needing to learn from other disciplines seem misplaced unless more context is provided. Eg What is it they do not know? What can other disciplines add to what psychologists know?

Response: We removed this sentence.

5. First question is about researcher's suitability to conduct the study. I think it will be clearer if the authors give examples from their experience in India and Thailand eg their

qualifications and its suitability for the study similar to what they do in discussing the other two questions. Additionally, I think it's a fallacy to imagine that participants in LMICs are 'disadvantaged' as the author mentions in this section. OR they can explain what they mean by disadvantage. Also explain what they mean by 'doing good'. Not all

Response: To address this comment, we changed the first sentence of this section to say: "Doing good cannot be uncritically assumed in global health or psychological research. It is essential that we, individuals in privileged positions, question our motivations for wanting to work with people who are disadvantaged (e.g., in terms of socio-economic status, ethnicity, or health)."

We also added more examples, e.g.: "For instance, when working with vulnerable community members, such as low-income older people, key concerns of participants will be how to meet basic health needs. Responding to these needs (e.g., by getting participants glasses so they can read the consent forms they are filling out) may go beyond the proposed research and funding guidelines but needs to be considered prior to planning the project."

Concerning the comment on "doing good": We mean ensuring ethical involvement of LMICs participants to start with. This includes safeguarding their rights, dignity, and well-being throughout the research process. When considering contributions to the community that we are studying, it is essential to clarify whether the research question/potential benefits are determined by the researchers or collaboratively defined with input from the participants. Highlighting the importance of co-designing research with collaborators in LMICs could be particularly relevant, as it ensures that the benefits align with local needs and perspectives. The whole section basically describes what we mean by doing good. We have also removed the quotation marks.

Reviewer #2 (Remarks to the Author):

This paper reflects on and acknowledges the challenges associated with conducting psychological research in lower-and middle-income countries (LMICs), based on the authors' research experiences in Myanmar, Thailand and India. The authors then proceed to pose and answer three critical questions that they believe researchers seeking to conduct or conducting psychological research in LMICs should be aware of, and reflect upon, to avoid potential pitfalls, and being exploitative. The paper builds upon neo-colonial science/helicopter research literature, and makes important practical contribution in terms of conducting responsible, ethical and decolonial research, particularly those involving participants from the Majority World settings. The contribution will be beneficial to both researchers and practitioners, and has the potential to influence thinking in the field.

Response: We would like to thank Dr. Adjei for his positive evaluation and insightful comments.

There are a few minor concerns I wish to bring to the attention of the authors for consideration:

1. Line 30: Take another look at this sentence, the last part seems a bit awkward for me. Do you mean "...trying to contribute to capacity strengthening to (instead of 'and') benefit communities?"

Response: We changed the sentence to make it more readable: "...and benefitting communities".

2. Line 33: Check the first question again. The content of the response doesn't seem to align very well with the question posed here. It seems to me that the question should rather be "Why do I need to involve/include local partners in this particular research?" or something similar.

Response: We kindly disagree with this suggestion to change the first question, as we address the question why we, as individuals in the global north, should be conducting research in the global south (that may not even be relevant to local communities or should be done by someone from the global south).

3. Line 48: It may be helpful to qualify/define the relationship here. Genuine relationship is preferred. This is important because of the various shapes and forms of unethical parachute/helicopter research that seem to be gaining traction the the academy. E.g., what qualify as local involvement? For e.g., when privileged academics from Minority World contexts/Global North pre-design every aspect of a research project, and only seek a couple of local partners to include for purposes of box-checking, they may think they are engaging in participatory research, because it involves relationship with local partners. However, this may not qualify as ethical and genuine research collaboration/relationship.

Response: We agree with this comment and have changed the sentence to address the quality and timing of relationships: "Genuine relationships with local partners—whether these are academics, staff from non-governmental organisations, civil society organisations, community and religious leaders—should be established before implementing a project to set an essential foundation for participatory research with communities."

Line 52-53: Please clarify whether these are either or situations. For example, is it enough to provide monetary compensation to a local RA without consideration for authorship? Should we even move to (would you advocate) crediting local participants with authorship, given that, as academics, we benefit from their rich expertise and knowledge- what we call data?

Response: Thank you for suggesting clarification. We chose to say "and", as we strongly believe in including local collaborators in authorships. We have also added a prominent citation to back up our argument: Abimbola, S. The foreign gaze: authorship in academic global health. *BMJ Glob Health* 4, e002068 (2019).

Line 70: 'there' should be 'There'

Response: We changed this accordingly.

Line 75: 'formative' should be 'Formative'

Response: We changed this accordingly.

Line 92-93: "semi-democratic" Myanmar? Whose evaluative definition is this? You or the local partners/participants? What does it mean? I would avoid such value-laden and evaluative terms.

Response: We removed this term (although it's commonly used by people from Myanmar to describe the time of the NLD party's rule between 2011-2021, where the military still had 25% of the seats in parliament and were not elected).

Line 147: The thought about spending enough time in an LMIC where research is to be conducted, preferably before co-designing and implementing a project, is a good point-should be mentioned earlier as well (see comment 3, line 48). Do you have ideas about how much time is 'enough', based on your own individual experiences?

Response: This question is challenging to answer definitively, as the appropriate amount of time spent will depend on the context, research question, and chosen methodology. For instance, field studies typically require more time compared to online survey studies. Although it may not be realistic for funders to spend several months at a time in a certain country, it is possible to build long-term relationships within the country over time, moving from small grants to co-develop ideas/build understanding to incrementally larger projects. We want to avoid being overly prescriptive by specifying a set duration, which is why we did not give an estimated time frame in the article. What is most important is avoiding helicopter research, where researchers engage only superficially. A balanced approach would be to collaborate with local partners and stakeholders to determine what constitutes sufficient time for meaningful engagement and accurate understanding within the specific context.

Regards

Stephen Baffour Adjei

Reviewer #3 (Remarks to the Author):

The manuscript addresses the ethical and practical challenges of conducting psychological research in low- and middle-income countries (LMICs), specifically Myanmar, Thailand, and India. The authors summarize key questions and concerns that researchers should consider in these contexts, emphasizing the importance of ethical reflexivity, community engagement, and culturally sensitive methodologies.

While the manuscript raises relevant points, much of the content overlaps with established ethical frameworks, such as the TRUST Code, and existing guidelines from various journals, including Nature. The repetition of familiar ethical principles may lead to a perceived lack of

novelty. The authors state in the introduction, “Here, we summarize key questions and concerns that we wish someone would have told us before conducting our research in Myanmar, Thailand, and India.” As readers, we expect to learn firsthand information from their experiences in different LMIC settings. However, upon reviewing the text, we find that much of the content aligns with well-established ethical frameworks. The authors provide some examples from their experiences, which is commendable. However, I believe they could enhance their contribution by offering more detailed accounts of specific challenges faced during their research in the three countries, particularly how these experiences shaped their methodologies and ethical considerations. This could provide valuable insights for researchers encountering similar situations.

Response: Thank you for this suggestion. We have added more examples to the commentary.

1. “For instance, when working with vulnerable community members, such as low-income older people, key concerns of participants will be how to meet basic health needs. Responding to these needs (e.g., by getting participants glasses so they can read the consent forms they are filling out) may go beyond the proposed research and funding guidelines but needs to be considered prior to planning the project.”
2. “In Southeast Asia, a common concept termed *kreng jai* (เกรงใจ) in Thai (anade အားနာတယ်) in Burmese, *phiền* in Vietnamese), which means to not want to burden others (e.g., by sharing personal problems and making others feel bad), prevented people from opening up to others in focus group discussions⁷. Hence, other research methods that involve ethnographic approaches to establish trusting relationships with participants may be more adequate in certain contexts.”
3. “It’s important to remember that benefits do not just come from research outputs; in Ethiopia, the most tangible impacts of Rosie’s work have been from contributions to mentoring and training, supporting early career researchers to successfully lead and publish peer-reviewed publications, apply for PhD fellowships, and transition into academic roles.”
4. (see next comment)

Furthermore, including a critique of existing ethical guidelines, including the TRUST Code, could illuminate areas needing improvement and spark a dialogue on how these codes can be adapted for diverse contexts. Overall, while this manuscript serves as a valuable starting point, it has the potential to offer more original insights for researchers in the field.

Response: We have added the need to adapt the TRUST code and illustrated it with an example from our field work: “However, the TRUST code also needs to be adapted to account for unique challenges that emerge when working in countries where there is very limited existing research guidelines. When Samia began her qualitative work on loneliness in Myanmar in 2019, following almost 50 years of military rule, one of her participants had security concerns after the interview as he had spent several years in the military himself (which led to deleting the interview recording). Learning from this experience, Samia made sure to explicitly promise not to discuss the political situation in the country when introducing her participatory project with Myanmar migrants in Thailand in 2023. By

ensuring on-going communication with participants, she was able to put them at ease and build trust, which in turn made them more open to contribute. This example illustrates how even psychological concepts such as loneliness can be perceived as political in a conflict-affected setting, requiring extra sensitivity in the research endeavour.”

We have also added some more key take-home messages from what we have learnt about doing work in LMICs in the conclusion: “Working in LMICs has made us actively reflect on how we can use our privileged role as researchers from HICs (or affiliated with institutions in HICs) to make a positive impact for the communities we work with, including in HIC contexts. It is relatively easy as a researcher in the Global North to become successful by branching out into more global research, justifying this as altruism, a motivation to address inequity. Over time and with increased knowledge of the historical, economic, and social context in which this takes place, it has become a much bigger challenge to explain why me, why this work? We have a shared recognition of how much we have personally gained from the kindness and openness of others to collaborate and share their knowledge and experiences. Whilst the impact of research can be difficult to ascertain, we hope that our shared approach—reflexive, flexible, focussed upon the relevance of our work (and ways of working)—has led to incremental reciprocal benefits for the people we’ve worked with, including mutual learning, skills and career development.”

We thank the author for their positive evaluation and helpful comments.